# Relationship between Dose Prescription Methods and Local Control Rate in Stereotactic Body Radiotherapy for Early Stage Non-Small-Cell Lung Cancer: Systematic Review and Meta-Analysis

**DOI:** 10.3390/cancers14153815

**Published:** 2022-08-05

**Authors:** Takahisa Eriguchi, Atsuya Takeda, Takafumi Nemoto, Yuichiro Tsurugai, Naoko Sanuki, Yudai Tateishi, Yuichi Kibe, Takeshi Akiba, Mari Inoue, Kengo Nagashima, Nobuyuki Horita

**Affiliations:** 1Radiation Oncology Center, Ofuna Chuo Hospital, Kamakura 247-0056, Japan; 2Department of Radiation Oncology, Keio University Hospital, Shinjuku, Tokyo 160-8582, Japan; 3Department of Radiation Oncology and Image-Applied Therapy, Kyoto University Hospital, Kyoto 606-8507, Japan; 4Department of Radiation Oncology, Tokai University Hachioji Hospital, Hachioji 192-0032, Japan; 5Department of Respiratory Medicine, Ofuna Chuo Hospital, Kamakura 247-0056, Japan; 6Biostatistics Unit, Clinical and Translational Research Center, Keio University Hospital, Shinjuku, Tokyo 160-8582, Japan; 7Chemotherapy Center, Yokohama City University Hospital, Yokohama 236-0004, Japan

**Keywords:** non-small-cell lung cancer, stereotactic body radiotherapy, dose prescription, biologically effective dose, local control

## Abstract

**Simple Summary:**

Stereotactic body radiotherapy (SBRT) is a standard treatment for inoperable early stage non-small-cell lung cancer (ES-NSCLC), and good local control and low toxicity rates have been reported. However, the optimal dose prescription method remains unclear. Variations in dose prescription methods make it difficult to properly compare the outcomes of SBRT for ES-NSCLC with those of previously published studies. Therefore, in this study, we conducted a comprehensive search of the published literature on the therapeutic results of SBRT for ES-NSCLC to summarize the results and clarify the relationship between LC and dose prescription methods. In our results, the central biologically effective dose of the planning target volume was most correlated with 3-year local control rates. A comparison against a standardized central biologically effective dose would show more definite outcomes of SBRT for ES-NSCLC and would help to strengthen its use in the treatment for ES-NSCLC.

**Abstract:**

Variations in dose prescription methods in stereotactic body radiotherapy (SBRT) for early stage non-small-cell lung cancer (ES-NSCLC) make it difficult to properly compare the outcomes of published studies. We conducted a comprehensive search of the published literature to summarize the outcomes by discerning the relationship between local control (LC) and dose prescription sites. We systematically searched PubMed to identify observational studies reporting LC after SBRT for peripheral ES-NSCLC. The correlations between LC and four types of biologically effective doses (BED) were evaluated, which were calculated from nominal, central, and peripheral prescription points and, from those, the average BED. To evaluate information on SBRT for peripheral ES-NSCLC, 188 studies were analyzed. The number of relevant articles increased over time. The use of an inhomogeneity correction was mentioned in less than half of the articles, even among the most recent. To evaluate the relationship between the four BEDs and LC, 33 studies were analyzed. Univariate meta-regression revealed that only the central BED significantly correlated with the 3-year LC of SBRT for ES-NSCLC (*p* = 0.03). As a limitation, tumor volume, which might affect the results of this study, could not be considered due to a lack of data. In conclusion, the central dose prescription is appropriate for evaluating the correlation between the dose and LC of SBRT for ES-NSCLC. The standardization of SBRT dose prescriptions is desirable.

## 1. Introduction

Stereotactic body radiotherapy (SBRT) is a standard treatment for inoperable early-stage non-small-cell lung cancer (ES-NSCLC), and good local control (LC) and low toxicity rates have been reported for SBRT [1,2,3]. With these positive results, the use of SBRT for NSCLC is annually increasing [4,5,6]. SBRT has rapidly evolved owing to the application of intensity-modulated radiotherapy, improved accuracy of target positioning, and accumulation of knowledge [7]. However, the optimal dose prescription method remains unclear. Dose prescription methods currently include the total dose, dose fractionation, site of dose prescription, and inhomogeneity of the dose distribution. In Japan, 48 Gy in four fractions from the isocenter has been most often prescribed [8] following a prospective clinical trial of SBRT for stage I NSCLC (JCOG0403) [3], while 54–60 Gy in three fractions has been most often prescribed at the periphery of the planning target volume (PTV) in the USA and Europe [9,10,11]. However, even in the same country, treatment planning varies. Giglioli et al. compared SBRT plans in 26 Italian institutions for stage I NSCLC with instructions that covered at least 95% of the PTV volume with 95% of the prescription dose [12]. Their results showed that the mean PTV dose greatly varied from 105 to 161 Gy as the equivalent uniform dose. These variations in dose prescription methods make it difficult to properly compare the outcomes of SBRT for ES-NSCLC among the published studies.

Therefore, in this study, we conducted a comprehensive search of the published literature on the therapeutic results of SBRT for ES-NSCLC to summarize the results and clarify the relationship between LC and dose prescription methods.

## 2. Materials and Methods

### 2.1. Study Overview

This study was designed, conducted, and reported in accordance with the Meta-analyses of Observational Studies in Epidemiology statement (Appendix A) [13]. Ethical approval from the relevant institutional review board was waived because we had no direct involvement with patients or original data. The study protocol was registered with the University Hospital Medical Information Network Center Clinical Trials Registry (UMIN-CTR, Tokyo, Japan) as UMIN000045014 [14].

### 2.2. Study Selection and Design

Our original plan was to include observational studies that reported the LC of SBRT for peripheral ES-NSCLC. The included articles were published as complete reports. Non-English studies were excluded from this study. During the initial screening, studies on diagnostic radiology, radiation physics, and radiobiology were excluded. Review articles and case reports were also excluded. We also excluded articles that did not mention LC at all. To analyze the relationship between the BED and LC, we selected studies with a median follow-up of at least 24 months and at least 25 patients.

### 2.3. Study Selection-Patients

Patients with NSCLC who were treated with SBRT were included in this study. Studies with a small proportion of central or T3 tumors were permitted. Studies involving patients with metastatic lung tumors were excluded. Studies limited to the specific characteristics of the tumor, such as pathology, ground-glass opacity, postoperative recurrence, or T3 or T4 tumors, were excluded. Studies with specific patient characteristics, such as chronic obstructive pulmonary disease or elderly patients, were permitted.

### 2.4. Study Selection-Treatment

SBRT or stereotactic ablative radiotherapy (SABR) using photon beam radiotherapy delivered within 10 fractions was included. Concurrent chemotherapy was not permitted. The inclusion and exclusion criteria of this study are shown in Appendix A.

### 2.5. Study Search

In this systematic review, we searched PubMed on 1 August 2021. The search strategy according to the PRISMA statement is shown in Figure 1, and the search formulae are shown in Appendix A. We manually searched the references lists of the included and reviewed articles.

### 2.6. Study Quality Assessment

The quality of each study was assessed using the Newcastle-Ottawa Scale (NOS) [15]. These scores are presented in Appendix A.

### 2.7. Outcomes

The correlation between the LC and four types of biologically effective doses (BEDs), which were calculated from the nominal prescription dose, minimum dose of PTV, maximum dose of PTV, and average dose of minimum and maximum dose of PTV, were evaluated.

### 2.8. Data Extraction

The articles identified in the database search were screened based on title and abstract information, and two authors (T.E. and A.T.) independently read and scrutinized the full text, and the necessary data were extracted. Discrepancies were resolved by consensus. From the screened studies, the following information was collected: first author, institution, year of publication, country, number of patients, median follow-up duration, dose fractionation, where the dose was prescribed, degrees of dose inhomogeneity in the PTV (% isodose), use of inhomogeneity correction, type of inhomogeneity correction, and adverse events. To analyze the correlation between the BED and LC, we extracted the 3-year LC rates and near-minimum or near-maximum doses of the PTV. The 3-year LC rate could be read on the Kaplan–Meier curve if the 3-year LC rate was not provided in the manuscript. If there were multiple prescription doses, the LC from each prescription dose was separately analyzed. When multiple prescription doses and LCs were reported, and they were not separately analyzed, the prescription dose applied for more than 80% of the patients was selected for the analysis. If multiple articles were published from the same institution, any duplication was checked, and the article with the highest number of cases was selected.

When multiple %-isodose levels for dose prescriptions were described, and the range did not exceed 10%, the maximum dose was calculated from the described median value of the percent isodose levels, if available. If the average value of the percent isodose level was available, the maximum dose was calculated from the average value of the percent isodose levels. Otherwise, the maximum dose was not calculated.

In two clinical trials, JCOG 0403 [3] and RTOG 0236 [9], doses were calculated using the old calculation algorithm. They were recalculated using the calculation algorithm with an inhomogeneity correction, and the dose corrections are reported [16,17]. Therefore, in those studies and in studies conducted using similar methods, the doses were corrected to the doses calculated with the calculation algorithm with inhomogeneity correction. Specifically, for a prescription dose of 48 Gy in four fractions from the isocenter of JCOG 0403 that Japanese institutions have commonly adopted, the peripheral dose of the PTV calculated with the superposition algorithm was revealed to be equivalent to 42 Gy [16]. Therefore, the peripheral dose was converted to 42 Gy in reports from Japanese institutions with this prescription and no inhomogeneity correction. Similarly, for reports whose prescription dose was the same as that in the RTOG 0236 trial of 60 Gy in three fractions [9], without inhomogeneity correction, the peripheral dose was set to 54 Gy [17].

A linear quadratic equation was used to calculate the BED: BED = *nd* [1+*d*/(*α*/*β*)], where *d* represents the dose per fraction, *n* is the number of fractions of SBRT, and *α*/*β* = 10, which was used for the analysis. From the information on dose prescriptions extracted from the screened articles, four kinds of BEDs (the nominal, central, peripheral, and average) were defined as follows: the nominal BED was calculated from the nominal prescription dose, which was the actual prescription dose described in each report; the central BED was calculated from near the maximum dose of PTV (PTV maximum dose, isocenter dose, or D2); and the peripheral BED was calculated from near the minimum dose of PTV (PTV peripheral dose, or D95–D98 of PTV). When both central and peripheral BEDs were available, the average BED was calculated as the average of the central and peripheral BEDs.

### 2.9. Statistics

Univariate and multivariate meta-regression analysis was performed, and the weighted Spearman’s rank correlation coefficient was calculated to analyze the relationship between the BED and LC. Statistical analyses were performed using OpenMeta [Analyst] [18] and Python. Statistical significance was defined as *p* < 0.05.

## 3. Results

We identified 1350 articles from database searches, and 14 articles more were included after a manual search. After excluding reviews, letters, editorials, case reports and series, and laboratory studies, 452 studies remained for the eligibility assessment. After the eligibility assessment, 188 studies were selected for the qualitative synthesis. Figure 2a–d shows the number of articles in each region over time, transition to the use of inhomogeneity correction, rates of fractionation, and dose heterogeneity in each region. Figure 2a shows that the number of published articles increased with time. As for the rates of increase in different regions, those from the USA and Canada increased more over time compared with those from Europe and East Asia. Regarding inhomogeneity corrections (Figure 2b), the use of an inhomogeneity correction was mentioned in less than half of the articles, even those most recent. Correction methods were described in only approximately 20% of the articles. Among the articles most recently written (2015–present), advanced heterogeneity correction methods (Monte Carlo and Acuros XB) were used more often; however, pencil beams were still used. As for rates of fractionation and dose heterogeneity (Figure 2c,d), regional disparities were found: in Europe and the USA and Canada, three fractions and high dose heterogeneity in the PTV were often used; on the other hand, in East Asia, four fractions and low dose heterogeneity in the PTV were often used.

To evaluate the relationship between the BED and LC, 33 studies, which included 3747 patients, were analyzed. The characteristics of these 33 studies are shown in Table 1. Among these 33 studies, both the central and peripheral BEDs were obtained in 24 studies. The central or peripheral BEDs were only obtained in one or eight studies, respectively.

On univariate meta-regression analysis, we found that the coefficients between the 3-year LC and nominal, central, peripheral and average BEDs were 2.6 × 10^−3^ (95% confidence interval (CI): −4.5 × 10^−3^–9.7 × 10^−3^; *p* = 0.48), 3.6 × 10^−3^ (95% CI: 3.0 × 10^−3^–6.8 × 10^−3^; *p* = 0.03), 3.7 × 10^−3^ (95% CI: −3.0 × 10^−3^–10.4 × 10^−3^; *p* = 0.28), and 4.4 × 10^−3^ (95% CI: −0.6 × 10^−3^–9.4 × 10^−3^; *p* = 0.08), respectively. Only the central BED significantly correlated with the 3-year LC among the four coefficients (Table 2 and Figure 3). A central BED of 150 Gy resulted in an LC of 90%, and a 30 Gy increase was expected to improve the 3-year LC rate by approximately 1%. In a multivariate analysis incorporating the percentage of T1 tumor, patient age, median follow-up period, and publication year, the central and average BEDs were significantly associated with the 3-year LC (*p* < 0.01 and 0.02, respectively) (Table 2 and Figure 3). The weighted Spearman’s rank correlation coefficients were significant between the 3-year LC and central and average BEDs (*p* = 0.01 and 0.03, respectively) (Appendix A). 

Among the 5181 patients, 210 (4.1%) had grade 3 or higher pulmonary toxicities. Grade 3 or higher toxicities other than pulmonary toxicities occurred in 1.2% of patients. Those toxicities rates did not correlate with any type of BED.

## 4. Discussion

This is the first systematic review regarding the site of dose prescription correlating with the LC rates in SBRT for ES-NSCLC. This study showed that the central BED was significantly but gradually correlated with the LC in the range of 100–300 Gy, and that this correlated with an LC better than the BED at the PTV periphery.

It is essential to determine the dose that is sufficient to achieve excellent local control in the treatment of SBRT for ES-NSCLC. However, this has not been sufficiently or systematically investigated. If at all, it was investigated with an undefined site of dose prescription. Onishi et al. reported that the LC was better with a BED of 100 Gy or more compared with less than 100 Gy [50]. Since then, a BED of 100 Gy has often been used as the threshold when discussing the correlation between the BED and LC. However, the original study was performed based on the nominal prescription dose, regardless of whether the sites of dose prescription varied among institutions. Zhang et al. conducted a meta-analysis to clarify the relationship between the nominal BED and the outcomes of SBRT for NSCLC [51]. They failed to show a significant correlation between the nominal BED and 3-year LC. This systematic review did not find a correlation either.

Several studies have investigated the correlation between the LC and the dose of the specific prescription site rather than only the nominal dose. The definition of dose prescription for SBRT planning is mainly divided into three categories: central prescription in PTV, which includes the maximum dose point, isocenter, and D2; peripheral prescription in PTV, which includes the peripheral dose and D90–99 in PTV; and middle prescription, which includes the mean dose and D50. As for the central prescription in PTV, some studies showed that the maximum BEDs of PTV were indices correlated with the LC [52,53,54]. The current systematic review also showed a significant correlation and found a gradual positive correlation between the central BED and LC, with a 30 Gy increase expected to improve the LC rate by 1%, and a central BED of 150 Gy resulted in an LC of 90%. Furthermore, Tateishi et al. reported that SBRT with a high maximum BED significantly improved the LC and OS [52]. Regarding middle prescriptions, some studies revealed a correlation between the mean dose of the PTV and LC [54,55,56]. Klement et al. analyzed a large database [54] and reported that the average BED of near-minimum and near-maximum BEDs was better correlated with tumor control probability (TCP) than either near-maximum or near-minimum BEDs. In contrast, the peripheral prescription was reported to have no significant relationship with the LC [57].

The prescribed dose to a specific location alone cannot fully explain the dose distribution in the PTV, although it plays a relevant role to some extent. In this study, we examined nominal, central, peripheral, and average BEDs (the average was calculated as the average of the central and peripheral BEDs) and found that the central BED was significantly correlated with the LC on univariate and multivariate meta-regression analyses and the weighted Spearman rank correlation coefficient. This probably is because the central BED reflects the true gross tumor volume dose and because the peripheral BED does not reflect this dose but only the marginal and lowest doses in the PTV. Additionally, even if the peripheral dose was the same, the tumor dose near the center of the PTV significantly varied with different isodose levels. The average BED was examined as one of the indices in this study because it was easy to extract from articles on SBRT for ES-NSCLC. However, the average dose was less strongly correlated with the LC than with the central dose. This may be because the correlation was weakened by averaging with the uncorrelated peripheral dose, or it may be because the dose distribution inside the PTV was not reflected. In this regard, the BED of D50 appears to be a better index than the average BED for determining the dose distribution inside the target volume. Even if D2 and D98 are the same in planning, D50 greatly differs between dose distributions with a steep summit and a flat top. The D50 proposed by ICRU91 [58] is simple and is reasonable for roughly determining the dose distribution, in addition to D2 and D98.

In dose calculation, the use or nonuse of inhomogeneity correction can cause considerable differences, especially in the periphery of the PTV, which is the boundary between the tumor and lung tissue densities. In the ICRU report 91, it is recommended to report the details of the treatment delivery software, such as grid size, algorithm, inhomogeneity correction, and specification of dose to water versus dose to tissue, in addition to the treatment planning system [58]. Commercially available treatment planning systems are classified by the type of inhomogeneity-corrected dose calculation algorithms from type A to type C. Type A includes all pencil beam convolution algorithms (PB) and does not consider changes in electron transport. Type B, which includes collapsed-cone convolutions (CCC) and anisotropy analysis algorithms (AAA), takes electron transport into account. Type C, including Monte Carlo (MC) and Acuros XB, considers the physics generating the dose absorption process [59]. A study [60,61] comparing the dose distributions calculated using several calculation algorithms revealed that the type-A algorithm overestimates the target dose; Latifi et al. compared the LC of NSCLC patients treated with SBRT calculated using two different dose calculation algorithms (PB and CCC algorithms). They revealed a significantly higher recurrence rate in the PB group (hazard ratio 3.4, 95% confidence interval: 1.18–9.83) [62]. Ohri et al. analyzed 928 patients who received SBRT for stage I NSCLC according to the calculation algorithm. In a multivariable Cox model adjusted for the tumor diameter and BED, the use of the PB algorithm was associated with an increased risk of local recurrence (hazard ratio, 2.39; 95% confidence interval, 1.08–5.29; *p* = 0.032) [63]. Despite these reports and the recommendation of ICRU report 91, half of the reports did not mention the inhomogeneity correction, even in the most recent, as can be seen in Figure 2b. Even in reports that used an inhomogeneity correction, 32% did not describe the details of the inhomogeneity correction used.

The fitting model for analyzing the relationship between different doses or fractions and the LC is controversial, especially in hypofractionated and large-fraction-dose SBRT. Because the linear-quadratic (LQ) model has been considered unsuitable for a large fraction dose, a variety of alternative models that have improved fitting in the high-dose region have been proposed in recent years [64,65,66]. In the conceptual phase of this systematic review, we attempted to use the LQ model or alternative models for TCP fitting, which represents the tumor control rate for the dose in the shape of a sigmoid curve. However, we used linear approximation to analyze the relationship between the BED and LC because TCP fitting with a sigmoid curve was unrealistic, given that the range of the BEDs in SBRT was limited to the area around the shoulder of the right sigmoid curve [67], despite its disadvantage of exceeding one for an extremely high BED.

This study has some limitations. It is unclear how a mixture of reports with and without heterogeneity correction affects the results. Future studies should comply with ICRU 91 to better estimate the effect of the dose and its prescription method on tumor control. Tumor volume is an important factor but was not included in the meta-regression analysis because only 11 studies provided tumor diameter or tumor volume. Other factors that might affect the LC, such as tumor location, pathology, policy to construct PTV, and machines, were not considered. The 95% CIs for meta-regression analysis were not calculated due to a lack of functionality in the analysis software.

## 5. Conclusions

The central dose prescription is appropriate for evaluating the correlation between the dose and LC of SBRT for ES-NSCLC. A central BED of 150 Gy resulted in an LC of 90%, and a 30 Gy increase in the central BED is expected to improve the LC rate by approximately 1%. The description of dose prescription should be based on the recommendation of ICRU report 91. The standardization of the descriptions of SBRT dose prescriptions is desirable.

## Figures and Tables

**Figure 1 cancers-14-03815-f001:**
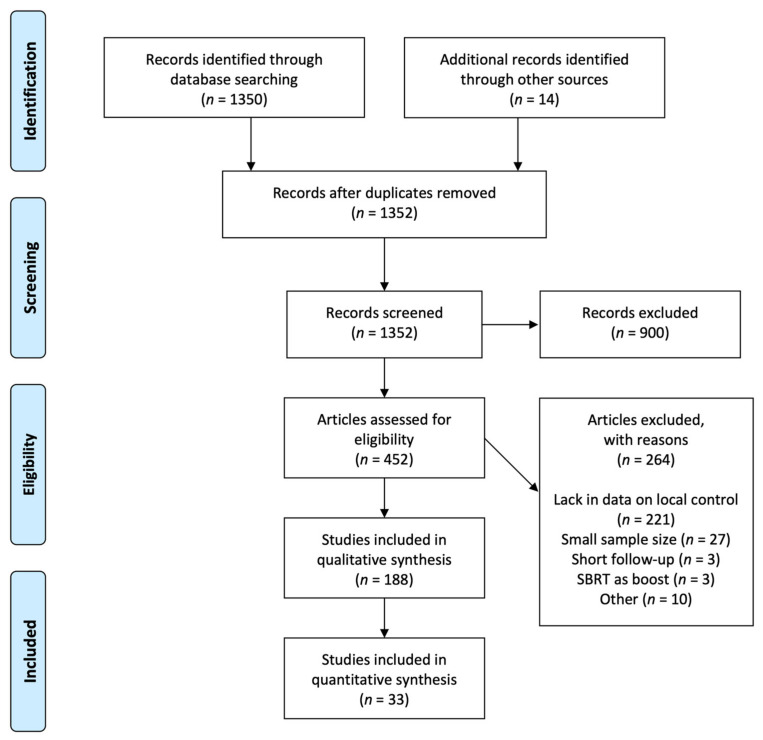
Flow chart for study selection.

**Figure 2 cancers-14-03815-f002:**
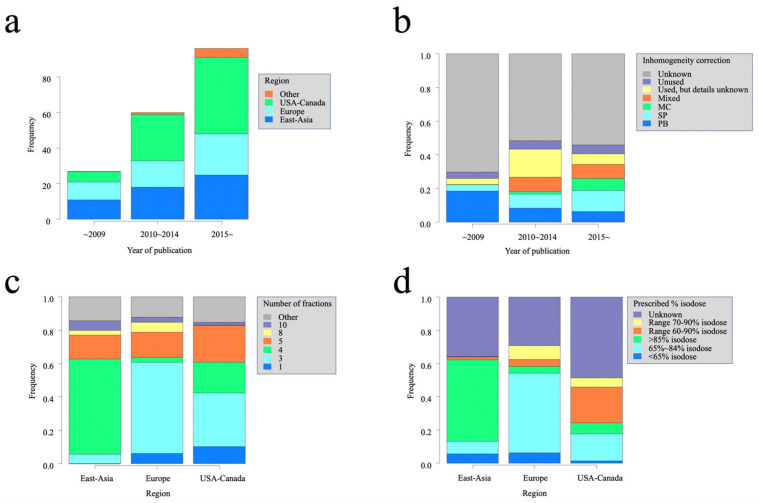
Characteristics of SBRT by region and year of publication. (**a**) Number of reports by year of publication, (**b**) use of inhomogeneity correction by year of publication, (**c**) number of fractions by region, and (**d**) prescribed percent isodose by region. Abbreviations: MC = Monte Carlo or Acuros XB; SP = superposition, anisotropic analytical algorithm, or collapsed cone convolution; PB = pencil beam convolution.

**Figure 3 cancers-14-03815-f003:**
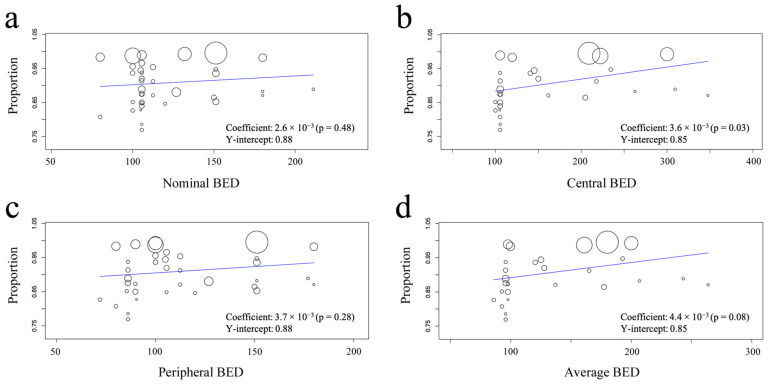
Univariate meta-regression analysis of 3-year local control rate according to (**a**) nominal, (**b**) central, (**c**) peripheral, and (**d**) average BEDs.

**Table 1 cancers-14-03815-t001:** Characteristics of included studies.

1st Author	Year	Country	*n*	Median f/u (Month)	Nominal/Peripheral/Central Dose (Gy)	Fraction	Nominal/Peripheral/Central/Average BED (Gy)	3-Year LC (%)
Nagata [19]	2005	Japan	32	30	48/42.0/48.0	4	105.6/86.1/105.6/95.9	95
Salazar [20]	2008	USA	60	38	40/40.0/52.0	4	80.0/80.0/119.6/99.8	98
Takeda [21]	2009	Japan	121	31	50/50.0/62.5	5	100.0/100.0/140.6/120.3	94
Fakiris [22]	2009	USA	34	50	60/54.0/75.0	3	180.0/151.2/262.5/206.9	88
Fakiris	2009	USA	36	50	66/59.4/82.5	3	211.2/177.0/309.4/243.2	88
Brown [23]	2009	USA	31	28	60/60.0/88.2	3	180.0/180.0/347.5/263.8	85.8
Ricardi [24]	2010	Italy	62	28	45/45.0/56.3	3	112.5/112.5/161.7/137.1	87.8
Grills [25]	2010	USA	58	30	48/48.0/–	4	105.6/105.6/–/–	96
Timmerman [9]	2010	USA	55	34	60/60.0/–	3	180.0/180.0/–/–	97.6
Shirata [26]	2012	Japan	45	30	48/43.2/48.0	4	105.6/89.9/105.6/97.7	100
Shirata	2012	Japan	29	30	60/54.0/60.0	8	105.0/90.5/105.0/97.7	82.1
Inoue [27]	2013	Japan	109	25	40/40.0/48.0	4	80.0/80.0/105.6/92.8	81
Suzuki [28]	2014	Japan	162	39	48/–/48.0	4	105.6/–/105.6/–	84
Hamaji [29]	2015	Japan	104	43	48/42.0/48.0	4	105.6/86.1/105.6/95.9	77
Lindberg [30]	2015	Sweden	57	42	45/45.0/67.2	3	112.5/112.5/217.7/165.1	92
Shibamoto [31]	2015	Japan	180	53	48/43.2/48.0	4	105.6/89.9/105.6/97.7	85
Hayashi [32]	2015	Japan	81	29	48/42.0/48.0	4	105.6/86.1/105.6/95.9	91.8
Nagata [3]	2015	Japan	169	56	48/42.0/48.0	4	105.6/86.1/105.6/95.9	87.6
Shaverdian [33]	2016	USA	110	29	54/54.0/65.6	3	151.2/151.2/209.0/180.1	100
Tsurugai [34]	2016	Japan	234	35	48/42.0/48.0	4	105.6/86.1/105.6/95.9	89
Navarro-Martin [35]	2016	Spain	38	48	54/54.0/70.2	4	151.2/151.2/234.5/192.8	94
Mancini [36]	2016	USA	251	36	54/54.0/–	3	151.2/151.2/–/–	85.3
Aoki [37]	2016	Japan	74	25	50/45.0/50.0	5	100.0/85.5/100.0/92.8	85
Sun [38]	2017	USA	65	86	50/50.0/–	4	112.5/112.5/–/–	95
Miyakawa [39]	2017	Japan	71	44	48/43.2/48.0	4	105.6/89.9/105.6/97.7	88
Lee [40]	2018	Korea	155	32	60/60.0/72.7	4	150.0/150.0/204.8/177.4	86.3
Raghavan [41]	2018	USA	140	39	54/54.0/–	3	151.2/151.2/–/–	93.4
Cummings [42]	2018	USA	65	24	30/30.0/–	1	120.0/120.0/–/–	84
Cummings	2018	USA	98	40	50/40.0/50.0	5	100.0/72.0/100.0/86.0	83
Karasawa [43]	2018	Japan	56	127	48/42.0/48.0	4	105.6/86.1/105.6/95.9	78.2
Menoux [44]	2018	U.K.	90	35	60/60.0/75.0	8	105.0/105.0/145.3/125.2	94
Tsurugai [45]	2019	Japan	157	28	50/50.0/83.3	5	100.0/100.0/222.1/161.0	99
Tsurugai	2019	Japan	66	28	60/50.0/100.0	5	132.0/100.0/300.0/200.0	100
Ball [46]	2019	Australia	66	31	48/48.0/–	4	105.6/105.6/–/–	85
Weiss [47]	2020	USA	100	32	48/48.0/60.0	4	105.6/105.6/150.0/127.8	92
Shu [48]	2020	China	68	46	50/50.0/–	5	100.0/100.0/–/–	95.6
Duvergé [49]	2021	France	418	41	54/54.0/–	4	126.9/126.9/–/–	88

Abbreviations: BED = biologically effective dose; 3-year LC = 3-year local control.

**Table 2 cancers-14-03815-t002:** Meta-regression analysis for 3-year local control and BED.

	UVA		MVA (+Percentage of T1 Tumor, Patient Age, f/u Period, Year of Publication)	
	Coefficient (95%CI) (×10^−3^)	*p*	Coefficient (95%CI) (×10^−^^3^)	*p*
Nominal BED	2.6 (−4.5–9.7)	0.48	2.9 (−6.7–12.6)	0.55
Central BED	3.6 (3.0–6.8)	0.03	5.4(1.6–9.1)	<0.01
Peripheral BED	3.7 (−3.0–10.4)	0.28	6.3 (–0.3–15.9)	0.20
Average BED	4.4 (−0.6–9.4)	0.08	7.4 (1.2–13.6)	0.02
Percentage of T1 tumor	−1.0 (−12.5–10.4)	0.86	–	–
Median patient age	−0.1 (−0.8–0.7)	0.88	–	–
Median f/u period	−0.1 (−0.2–0.1)	0.18	–	–
Year of publication	−0.1 (−0.6–0.4)	0.63	–	–

Abbreviations: BED = biologically effective dose; UVA = univariate analysis; MVA = multivariate analysis; f/u = follow-up.

## Data Availability

Data generated or analyzed during the study are available from the corresponding author by request.

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
