# Peer review of "Relationship between Dose Prescription Methods and Local Control Rate in Stereotactic Body Radiotherapy for Early Stage Non-Small-Cell Lung Cancer: Systematic Review and Meta-Analysis"

_cancers, 2022, doi:10.3390/cancers14153815_

Round 1

Reviewer 1 Report

Authors declare to provide a systematic review regarding the correlation between several physical aspects of dose prescription (nominal, central periphery and average) and local control after SBRT in NCLC.

I agree with the authors that the difference in prescription make difficult to compare results between different clinical studies. Anyway in my opinion the issue cannot be faced in a systematic review.

The manuscript is not clear and quite confusonary in the study design and most of all in reporting of results. If it is a systematic review a PICO methodology should be used and MeSH terms of the research should be provided. Inclusion and exclusion criteria should be detailed point by point. Why did you considered only peripheral tumors? SBRT is delivered also on ES-NSCLC centrally located.

Lines 90-94 is unclear. Paragraph 2.8 is brainy and difficult to comprehend by readers.

Fig.2a,b,c,d are completely lacking of interest.

At univariate analysis only central BED resulted significantly correlated with LC. Multivariate analysis is not useful in this case. Why did you provide multivariate analysis considering again the same variables of the UVA? (Table 2 and lines 204-205). It is a great mistake! Only significant variables at UVA should be tested in the MVA.

Lines 197-202 and Figure 3 is not clear! Coefficent per 30Gy is not explained. What does it mean? 

Author Response

Response to Reviewer 1

Authors declare to provide a systematic review regarding the correlation between several physical aspects of dose prescription (nominal, central periphery and average) and local control after SBRT in NCLC.

I agree with the authors that the difference in prescription make difficult to compare results between different clinical studies. Anyway in my opinion the issue cannot be faced in a systematic review.

The manuscript is not clear and quite confusonary in the study design and most of all in reporting of results. If it is a systematic review a PICO methodology should be used and MeSH terms of the research should be provided. Inclusion and exclusion criteria should be detailed point by point. Why did you considered only peripheral tumors? SBRT is delivered also on ES-NSCLC centrally located.

Thank you for reviewing this paper. We will make every effort to respond sincerely to your severe comments so that the paper will be accepted as a better paper.

Usually, PICO is established in systematic reviews, but since the design of this study is to evaluate the correlation between LC and four different BEDs, we do not believe that PICO is applicable. Mesh term is included within the supplemental search formula, so please refer there.

We have created Table S2 with inclusion and exclusion criteria.

It is true that SBRT is performed not only for peripheral lesions but also for central lesions. However, in the case of central lesions, the dose and fractionation differ greatly from peripheral lung cancer because of the presence of risk organs such as bronchi and pulmonary arteries. In such cases, the local control rate will depend more on the location of the organ than simply the prescribed dose. In SBRT for lung cancer, many reports have been limited to peripheral lesions, which are not affected by the presence of such risk organs. Therefore, in this study, the analysis was also limited to peripheral lesions.

Lines 90-94 is unclear. Paragraph 2.8 is brainy and difficult to comprehend by readers.

Thank you for your suggestion, I have rewritten Lines 90-94. I have also reviewed paragraph 2.8 and made it more concise.

Fig.2a,b,c,d are completely lacking of interest.

The main objective of this study is to determine whether the doses at the three dose prescription sites affect local control. The four analyses in Figure 2 illustrate the factors inherent in the subject. The analysis of 188 articles shows that the number of reports of SBRT has increased with age, that many of the articles, even in recent years, do not mention heterogeneity correction, and that the dose prescription (frequency, %isodose) varies with the region. We find it interesting that the analysis of the literature showed.

At univariate analysis only central BED resulted significantly correlated with LC. Multivariate analysis is not useful in this case. Why did you provide multivariate analysis considering again the same variables of the UVA? (Table 2 and lines 204-205). It is a great mistake! Only significant variables at UVA should be tested in the MVA.

We understand that there is a concept that only factors that are significant in univariate analysis are used in multivariate analysis. However, many papers have included clinically meaningful factors in multivariate analyses even if they were not significant in univariate analyses [1,2]. In this analysis, publication year and age were included in the multivariate analysis to exclude the possibility of a bias toward favorable results for central BED because the studies reporting higher local control for central BED might have been published in more recent year and younger patients were treated.

  1. Leeman, J.E.; Chen, Y.H.; Catalano, P.; Bredfeldt, J.; King, M.; Mouw, K.W.; D'Amico, A.V.; Orio, P.; Nguyen, P.L.; Martin, N. Radiation Dose to the Intraprostatic Urethra Correlates Strongly With Urinary Toxicity After Prostate Stereotactic Body Radiation Therapy: A Combined Analysis of 23 Prospective Clinical Trials. International journal of radiation oncology, biology, physics 2022, 112, 75-82, doi:10.1016/j.ijrobp.2021.06.037.
  2. Teo, M.T.W.; McParland, L.; Appelt, A.L.; Sebag-Montefiore, D. Phase 2 Neoadjuvant Treatment Intensification Trials in Rectal Cancer: A Systematic Review. International journal of radiation oncology, biology, physics 2018, 100, 146-158, doi:10.1016/j.ijrobp.2017.09.042.

Lines 197-202 and Figure 3 is not clear! Coefficent per 30Gy is not explained. What does it mean? 

“Coefficent per 30Gy” means the percentage increase in local control per 30 Gray increase in biological effective dose. However, it is difficult to understand, so we changed it to "coefficient per 1 Gy".

Reviewer 2 Report

I was asked to review a manuscript entitled, "Relationship between dose prescription methods and local control rate in stereotactic body radiotherapy for early-stage non-small cell lung cancer: systematic review and meta-analysis." 

I think this is an important meta-analysis because there is great confusion over dosing for stereotactic treatment of lung cancer despite thousands of publications.  The authors bring up some very interesting points such as the dosing differences between US/Canada and East Asia as well as the fact that less than half the articles mention whether they used inhomogeneity corrections.

My main concerns must be addressed are in figure 3 and table 2.  The coefficient for 30 Gray is not clear to me.  This must be specified in the material and methods section.  Does this coefficient for 30Gy actually mean the percentage increase in local control per 30 Gray increase in biological effective dose?  In table 2 does age mean patient age?  Table 2 should definitely include follow-up time for the UVA and MVA. Average tumor size would also be a nice addition to table 2.

Minor concerns

1.  Can you please address why 900 records were excluded in figure 1?

2.  According to the New Castle-Ottawa scale, all of the studies are ranked a 5 and a 6 for moderate quality studies.  I think you should address this fact in your discussion section. Why do we not have high quality studies?

3.  Reference 59 is incomplete.

4. I think you should make a strong closing statement such as all studies on stereotactic body radiation therapy treatments of early-stage non-small cell lung cancer should include average peripheral dose, average percentage isodose prescription, average maximum tumor dose, inhomogeneity correction methodology, and physics calculation algorithm.

Author Response

Response to Reviewer 2

I was asked to review a manuscript entitled, "Relationship between dose prescription methods and local control rate in stereotactic body radiotherapy for early-stage non-small cell lung cancer: systematic review and meta-analysis." 

I think this is an important meta-analysis because there is great confusion over dosing for stereotactic treatment of lung cancer despite thousands of publications.  The authors bring up some very interesting points such as the dosing differences between US/Canada and East Asia as well as the fact that less than half the articles mention whether they used inhomogeneity corrections.

My main concerns must be addressed are in figure 3 and table 2.  The coefficient for 30 Gray is not clear to me.  This must be specified in the material and methods section.  Does this coefficient for 30Gy actually mean the percentage increase in local control per 30 Gray increase in biological effective dose?  In table 2 does age mean patient age?  Table 2 should definitely include follow-up time for the UVA and MVA. Average tumor size would also be a nice addition to table 2.

Thank you for your recognition of the importance of the subject matter of this research and for your review of this paper. I will make a sincere effort to respond to your comment and to get it accepted.

As you pointed out, "Coefficient for 30Gy" means "coefficient for 30Gy actually means the percentage increase in local control per 30 Gray increase in BED". However, it is difficult to understand, so we changed it to "coefficient per 1 Gy". We changed “age” to “patient age”. MVA was reanalyzed including the follow-up period. Average tumor size is an important factor, as you pointed out, but since only about 1/3 of the literature includes average tumor size or target volume, we used T-stage for multivariate analysis instead.

Minor concerns

  1. Can you please address why 900 records were excluded in figure 1?

During the initial screening, we exclude articles that are not clearly relevant to the study, e.g., articles on diagnostics, radiation physics, or radiobiology. Review articles and case reports are also excluded. We also exclude articles that do not mention LC at all. A sentence was added in 2.2. Study selection-design.

  1. According to the New Castle-Ottawa scale, all of the studies are ranked a 5 and a 6 for moderate quality studies.  I think you should address this fact in your discussion section. Why do we not have high quality studies?

“Selection of the non-exposed cohort" and "Comparability of cohorts on the basis of the design or analysis controlled for confounders" were scored 0 because there were no comparables in this study. Therefore, they were given a score of 0. As a result, the score is low. Although NOS scale is often used to assess study quality when conducting meta-analyses of non-randomised studies, the NOS does not fit well with the current study design, resulting in a low score.

  1. Reference 59 is incomplete.

Thank you for pointing this out. Reference 59 has been corrected.

  1. I think you should make a strong closing statement such as all studies on stereotactic body radiation therapy treatments of early-stage non-small cell lung cancer should include average peripheral dose, average percentage isodose prescription, average maximum tumor dose, inhomogeneity correction methodology, and physics calculation algorithm.

Your point is recommended in ICRU 91. We have added that " The description of dose prescription should be based on the recommendation of ICRU report 91” in conclusion.

Reviewer 3 Report

This is a very relevant and huge, comprehensive work looking at the relation between different BED calculations and local control in NSCLC patients treated with SBRT.

I have a few major remarks:

1. The local control rate is not only dependent on the radiation dose, but also on the volume of the tumor. I believe that this parameter should be included in the analysis. It may only be possible to use the T-stage as a proxy for individual volumes will not be available in literature. In that case, at least binning of volumes should be dose, e.g. tumors < 1 cm, 1-2 cm, etc.

2. There may be an intrinsic bias in the dose a tumor receives, its volume and its anatomical location. Indeed, smaller tumors that are located in an easy to irradiate place may receive higher doses, but these tumors may also have a better prognosis, independently of the radiation dose they are treated with.

3. Linear correlations are applied, whereas we know that most dose-response relations are sigmoidal. I would like to see non-linear statistics.

4. An estimation on the uncertainties on the dose-effect relation is needed.

Author Response

Response to Reviewer 3

This is a very relevant and huge, comprehensive work looking at the relation between different BED calculations and local control in NSCLC patients treated with SBRT.

I have a few major remarks:

  1. The local control rate is not only dependent on the radiation dose, but also on the volume of the tumor. I believe that this parameter should be included in the analysis. It may only be possible to use the T-stage as a proxy for individual volumes will not be available in literature. In that case, at least binning of volumes should be dose, e.g. tumors < 1 cm, 1-2 cm, etc.

 Thank you for your high evaluation and for reviewing this paper. I will respond sincerely to your comment and will make every effort to get it accepted.

As you pointed out, tumor volume is an important factor, but since only 11 of the studies provided average tumor size or tumor volume, we did not include it in meta-regression analysis. T-stages were obtained from almost all reports, so T-stages were used for multivariate analysis.

  1. There may be an intrinsic bias in the dose a tumor receives, its volume and its anatomical location. Indeed, smaller tumors that are located in an easy to irradiate place may receive higher doses, but these tumors may also have a better prognosis, independently of the radiation dose they are treated with. 

As you noted, tumor location and size can affect LC. Tumor size could affect LC, but as mentioned above, it was not fully available in current study. As for location, central lesions are excluded in the first place because they affect LC. Since there are only a few papers that describe the detailed location of the tumor (only 5 studies described lung lobes with tumors present), we have not been able to take that into account. Therefore, I have added a note to the limitation on this issue.

  1. Linear correlations are applied, whereas we know that most dose-response relations are sigmoidal. I would like to see non-linear statistics.

We also had a great deal of trouble deciding whether to use the sigmoid curve, and after analyzing similar review article published in Int J Radiat Oncol Biol Phys [1], we wrote a letter questioning the approximation to the sigmoid curve [2]. Because the SBRT has a high local control, we only have data near the right shoulder of the sigmoid curve. Therefore, when we tried to fit the data to the sigmoid curve, we found that we had to rely heavily on a small number of data with low local control rates, as in the review article. For this reason, in this study, we have ventured to approximate the data with linear regression. The above has already been discussed in detail in the discussion, so please refer to that.

  1. Lee, P.; Loo, B.W., Jr.; Biswas, T.; Ding, G.X.; El Naqa, I.M.; Jackson, A.; Kong, F.M.; LaCouture, T.; Miften, M.; Solberg, T.; et al. Local Control After Stereotactic Body Radiation Therapy for Stage I Non-Small Cell Lung Cancer. International journal of radiation oncology, biology, physics 2021, 110, 160-171, doi:10.1016/j.ijrobp.2019.03.045.
  2. Eriguchi, T.; Takeda, A.; Kimura, Y.; Sanuki, N. In Regard to Lee et al. International journal of radiation oncology, biology, physics 2021, 111, 1088-1089, doi:10.1016/j.ijrobp.2021.07.012.

  1. An estimation on the uncertainties on the dose-effect relation is needed.

The analysis software used in this study cannot calculate 95% CI. We have indicated this in the Limitation section.

Round 2

Reviewer 1 Report

I'm sorry but my opinion doesn't change after authors' replies as my too many major concerns raised after revision.

This is not a systematic review neither a metanalysis and the arguement doesn't suit for both of them. The issue could be interesting but it should be exposed in a clearer manner. Also, this is a very specialistic argument so it should be submitted to a journal specialized in radiation oncology.

I do advise to reformat the manuscript to make it easier to read 

Reviewer 2 Report

The author comments and improvements have made the manuscript acceptable

Reviewer 3 Report

I would like that a sentence is added in the abstract that the tumor volume, which is of importance for any outcome, could not be taken into account because of the lack of data.

This is mentioned in the discussion, but as most people only read abstracts, this important caveat should be mentioned in the abstract as well.

Otherwise, the revision is adequate.
